# Tricyclic Antidepressant Use and Risk of Fractures: A Meta-Analysis of Cohort Studies through the Use of both Frequentist and Bayesian Approaches

**DOI:** 10.3390/jcm9082584

**Published:** 2020-08-10

**Authors:** Qing Wu, Yingke Xu, Yueyang Bao, Jovan Alvarez, Mikee Lianne Gonzales

**Affiliations:** 1Nevada Institute of Personalized Medicine, University of Nevada, Las Vegas, NV 89154, USA; yingke.xu@unlv.edu (Y.X.); baoy9@mcmaster.ca (Y.B.); araioj@unlv.nevada.edu (J.A.); gonzam2@unlv.nevada.edu (M.L.G.); 2Department of Epidemiology and Biostatistics, School of Public Health, University of Nevada, Las Vegas, NV 89154, USA; 3Department of Biology, McMaster University, Hamilton, ON L8S 4L8, Canada; 4Department of Biology, School of life sciences, University of Nevada, Las Vegas, NV 89154, USA

**Keywords:** fractures, tricyclic antidepressants, bone density, depression, osteoporosis, Bayesian meta-analysis

## Abstract

Background: Research findings regarding the association between tricyclic antidepressant (TCA) treatment and the risk of fracture are not consistent; we aimed to assess whether people who take TCAs are at an increased fracture risk. Methods: Relevant studies published through June 2020 were identified through database searches of MEDLINE, EMBASE, Scopus, PsycINFO, ISI Web of Science, WorldCat Dissertations and Theses from each database’s inception, as well as through manual searches of relevant reference lists. Two researchers independently performed literature searches, study selection, data abstraction and study appraisal by using a standardized protocol. Frequentist and Bayesian hierarchical random-effects models were used for the analysis. The heterogeneity and publication bias were evaluated in this study. Results: Eight studies met the inclusion criteria. Overall, TCA use was associated with a significantly increased risk of fracture in both the frequentist approach (Risk Ratio (RR), 1.23; 95% CI, 1.06–1.42; *p* = 0.007) and the Bayesian method (RR, 1.24, 95% Credible Interval (CrI), 1.01–1.56). These results were consistent in multiple sensitivity and subgroup analyses. Significant heterogeneity was observed in the meta-analysis; however, no significant publication bias was detected. Conclusion: TCA medication may indicate an increased risk of fracture. TCA should be prescribed with caution in the clinic.

## 1. Introduction

With the rapid growth of aging populations on a global scale, osteoporotic fractures have become a critical public health concern worldwide. Globally, around 1.6 million hip fractures occur each year [1]. Additionally, fragility fractures lead to many adverse outcomes, including disability, decreased quality of life and excess mortality [2], especially in older individuals [3]. These consequences of osteoporotic fractures have already led to a significant increase in the social and economic burden on a global scale [4].

On the other hand, depression or depressive disorder is a prevalent mental illness, one that affects 17.3 million adults in the United States alone [5]. A widely employed treatment for depression is antidepressant medication, which is one of the most frequently prescribed medicines in Western countries [6]. Tricyclic antidepressants (TCAs) were developed in the 1950s to treat depression [7]. Currently, although TCAs have been increasingly replaced by selective serotonin reuptake inhibitors (SSRIs) and other new safer antidepressants, they are still a good choice for some patients whose depression has not responded to treatment with less toxic agents [8]. Previously we have demonstrated that depression is associated with a lower bone density [9] and higher fracture risk [10]. The link between SSRIs use and fracture risk was also examined extensively [11,12]. Many studies reported that TCA use is significantly associated with osteoporotic fractures [13,14,15,16,17,18,19,20,21,22,23,24]; however, other studies found that such an association was not significant [25,26,27,28]. As a result, we conducted a meta-analysis using both cohort and case-control studies to examine the association between TCA use and fractures [24]. However, the prior meta-analysis was conducted eight years ago, and thus several large eligible cohort studies undertaken in recent years were not included [27,28,29,30,31]. Therefore, an updated meta-analysis was warranted.

This meta-analysis aimed to quantitatively assess all eligible cohort studies that examined the effect of TCA use on fracture risk and to obtain a more comprehensive assessment and a more precise and accurate estimate about this effect. Besides the classical frequentist method, we also employed a Bayesian approach for this meta-analysis research, as Bayesian meta-analysis uses the probabilistic method and can help with clinically relevant decision-making when one is confronted with uncertainty.

## 2. Experimental Section

We conducted this meta-analysis research using the guidelines of Meta-analysis of Observational Studies in Epidemiology (MOOSE) [32,33]. The detailed statement of Preferred Reporting Items for Systematic Reviews and Meta-analyses (PRISMA) was referred to when applicable. The study objectives, primary outcomes, literature search strategy, inclusion and exclusion criteria, methods for study selection, data extraction, and data synthesis were defined in advance in the meta-analysis research protocol. In the protocol, we also prespecified the sensitivity and subgroup analyses we planned to conduct.

### 2.1. Search Strategy and Data Sources

We conducted a comprehensive literature search of MEDLINE (from 1946) using OVID. Without language restrictions, we used the search terms tricyclic antidepressants, antidepressant, amitriptyline, nortriptyline, protriptyline, imipramine, desipramine, doxepin, trimipramine, fractures, osteoporosis, osteopenia, bone density and bone (See search strategy in Appendix A). Using the same strategy, we also conducted literature searches of EMBASE (from 1988), PsycINFO (from 1806), SCOPUS (from 1960) and ISI Web of Science (from 1975). The above search terms were adapted for other database searches according to the syntax of each specific database. The last literature search was conducted on 22 May 2020. The literature search with MEDLINE was automatically updated to 24 June 2020with OVID Auto Alert. We also searched WorldCat Dissertations and Theses from its inception, as well as the proceedings of the International Osteoporosis Foundation World Conference on Osteoporosis and the conference abstracts of the American Society for Bone and Mineral Research from 2000 to 2020. An experienced librarian in health science was consulted during the literature search. Two investigators (Y.X. and Y.B.) independently examined reference lists from the original studies [23,26,27,28,29,30,31] and related meta-analyses and reviews [24,34,35,36,37].

### 2.2. Study Selection

During the initial screening phase, simplified inclusion criteria were used to screen relevant references: (1) human subjects, (2) fracture or bone mineral density (BMD) as the outcome and (3) TCA use as exposure. Each title and abstract of each article retrieved from the electronic search was independently reviewed by two investigators (Q.W. and Y.X.). Only those citations that both reviewers deemed irrelevant were excluded. References with a disagreement between the two reviewers were included for a further full review.

In the second phase of the study selection, the full content of each study obtained during the screening stage was reviewed and assessed. We included cohort studies that reported data on the subjects who had used TCAs and on the participants who were not exposed to antidepressants. The outcome variables were reported as either fractures, BMD or both. No clinical trial studies were found to be eligible. Cross-sectional studies were excluded from this meta-analysis. Using prespecified selection criteria and assessment protocols, two investigators (Q.W. and Y.X.) independently assessed the full content of each article in English and Chinese. Articles in other languages were reviewed and evaluated by additional investigators who had the corresponding multilingual expertise, using the same criteria and assessment protocol. Areas of disagreement or uncertainty were resolved by consensus. We only included studies that reported the hazard ratio (HR) or risk ratio (RR) of fractures, or/and the BMD change associated with TCA use. HR is broadly equivalent to RR [38]; thus, we approximated HRs as RRs. The agreement between investigators was evaluated using the κ statistic, a robust statistic for testing for interrater reliability [39]. The kappa value was calculated using the following formula:κ = [Pr(a) − Pr(e)]/[1 − Pr(e)],(1)
where Pr(a) is the actual observed agreement and Pr(e) is the expected agreement.

### 2.3. Study Appraisal

We evaluated the methodological quality of the included studies by using the Newcastle-Ottawa Scale [40]. As recommended by the MOOSE study team [33], the quality scores were not used as weights for the corresponding studies in the meta-analyses. Instead, the quality scores were used in a subgroup analysis (>7 vs. ≤7).

### 2.4. Data Abstraction

We used the data abstraction form modified from our previous meta-analysis [24]. Two investigators (Q.W. and Y.X.) independently abstracted all data through the use of the data abstraction form. No major disagreements or discrepancies arose between the two investigators; minor differences were resolved by rechecking the original reports and by discussion. We abstracted the following information from each study: the study characteristics (the names of authors, the year of the publication and the journal if applicable, and the country where the research was conducted), the study design, the study setting, the inclusion criteria, the sample size and duration of the follow-up, the participants’ characteristics (age, gender and race, if available), the outcomes (fractures or BMD change) and corresponding regions and ascertainment methods, the assessment of TCA use, the statistical analysis methods, and the estimates of the effect size (adjusted RR, HR, BMD change and their 95% CIs). For the overall pooled analysis, when original reports presented multiple estimates, we selected the effect size that adjusted for the most confounders, the estimate derived from the larger sample size, the estimate from current TCA users, and the estimate from an osteoporotic fracture, if applicable. Corresponding estimates from the subgroup analyses in the original studies were abstracted when appropriate. We did not contact the authors of the original studies because no additional information was required.

### 2.5. Statistical Analysis

We used the confounder-adjusted RR to measure the association between TCA use and fracture risk. HR was considered equivalent to RR. For studies that reported the estimates by subgroups only, the overall effect size was estimated by a meta-analysis of the reported subgroup’s estimates. To normalize the data distribution and to stabilize the variance, we transformed HRs or RRs into their natural logarithms [41] for the pooled meta-analysis. We derived the variance of the natural logarithm of the HR or RR from the corresponding 95% CIs provided by the original reports. To calculate the overall estimated RR and BMD change, we weighted each included study by the reciprocal of the corresponding variance reported in the original studies. Both frequentist and Bayesian hierarchical random-effects models were utilized for the synthesis analysis. In the frequentist meta-analysis, the DerSimonian-Laird method [42] was used to calculate the pooled RR and variance. In the Bayesian meta-analysis, Gaussian distribution with an unknown effect size (*θ_i_*) and known within-study variance $\delta_{i}^{2}\;$ was assumed for each log RR (denoted as *φ_i_*). The set of *θ_i_* across the original studies was also assumed to follow a Gaussian distribution, with an unknown mean (*μ*) and across-study variance (*τ*^2^), where *μ* was the estimate of the overall log RR and *τ*^2^ was a measure of the between-study variation. The prior distribution of *τ*^2^ was assumed to follow an improper uniform distribution, and the prior distribution for *τ*^2^ was assumed to be non-informative. The probabilities that TCA use increases fracture risk by more than 0%, 10% or 20% were estimated and reported.

To assess the robustness of our estimates, we conducted several prespecified sensitivity analyses. The effects of TCAs on fracture risk were calculated with different inclusion criteria, including using TCA as the primary exposure, using osteoporotic fractures as the outcome, and studies focusing on people younger than 65 years old. The Cochran Q statistic [43] and the Higgins index I^2^ were employed to assess heterogeneity [44]. We applied the random-effects model for this meta-analysis because of the observed heterogeneity between the original studies.

To determine whether demographic and clinical variables modified the effect of TCAs on fracture risk, we conducted several prespecified subgroup analyses. These variables included the anatomical site of the fracture, adjustment of fracture-related confounders (BMD, smoking, osteoporosis), year of publication and study location. As most studies did not specify race/ethnicity and others used multiethnic populations, a subgroup analysis for race/ethnicity was not performed. We conducted a cumulative meta-analysis by performing sequential random-effects pooling, beginning with the earliest qualified report. Each subsequent meta-analysis summarized all eligible reports in the preceding years. To demonstrate the effect of adding reports on the pooled effect size, we presented results chronologically in a forest plot. A multivariate meta-regression analysis was not performed because the number of eligible studies was small, and because some key variables, such as gender and race/ethnicity, were not available in some original reports.

To examine the potential for publication bias, we constructed a funnel plot by plotting RRs against their standard errors [45]. We also used the Begg and Mazumdar rank correlation test [46] to examine the significance of publication bias. Furthermore, we employed the trim-and-fill method to estimate and adjust for the potential effects that nonpublished studies might have had on the estimated effect size. We used R statistical software (Version 4.0, Core Team, Vienna, Austria) for the data analysis. A *p*-value of 0.05 or less was considered statistical significant.

## 3. Results

The study selection flow is illustrated in Figure 1. After removing duplicate references from different databases, we found a total of 6336 potential references. After the two investigators (Q.W. and Y.X.) screened titles and abstracts of all these references, 137 full-text research articles were retrieved and assessed for eligibility. The agreement between the two investigators was modest at this initial screening stage (κ = 0.70). After reviewing all full-text articles, eight studies with fracture data met the inclusion criteria. The agreement between the two investigators was good at this second stage (κ = 0.93). No related randomized controlled trials were eligible, and no studies with BMD outcomes were qualified. All eight included studies were published in English.

Table 1 showed participants’ characteristics and related information from the eight eligible cohort studies. Each included study was controlled for confounding effects of age and sex (if applicable). The mean follow-up period ranged from five to ten years. Of the eight studies, three were conducted only on persons aged ≥65 years [23,28,30], four of the eight were performed in Europe [26,27,29,31], three were conducted in North American [23,25,30], and one reported results from different regions [28].

Figure 2 shows the pooled RR estimated through the use of both frequentist and Bayesian approaches. It demonstrates the fracture risk (RR and 95% CI or CrI) associated with TCA treatment in each original study and all studies combined. Compared with patients who had not taken TCAs, those who had taken TCAs had an overall RR of 1.23 (95% CI, 1.06–1.42) in the frequentist approach and 1.24 (95% CrI, 1.01–1.56) when using the Bayesian method. The probabilities that TCA use increased fracture risk by more than 0%, 10% and 20% were 98%, 89% and 63%, respectively. Significant heterogeneity was observed among the eight studies in this meta-analysis, as the Cochran Q statistic was significant (*p* < 0.01), and the Higgins I^2^ index was 82%.

The estimated fracture risk associated with TCA use changed little when studies had different inclusion criteria (Table 2). For example, the overall estimated RR did not alter when only studies using TCA as the primary exposure were included (RR, 1.23; 95% CI, 1.05–1.43). The effect size decreased slightly to 1.20 (95% CI, 1.00–1.43) when studies that reported HR for the risk estimation were included. The effect size increased slightly to 1.37 (95% CI, 1.04–1.47) when studies with osteoporotic fractures as the outcome were included. After the exclusion of three studies that only included persons older than 65 years, the overall RR increased slightly to 1.25 (95% CI, 1.04–1.52). The cumulative meta-analysis (Figure 3) by the frequentist method demonstrated that the pooled RR associated with TCA use fluctuated over time; however, it was consistently significant since 2011, as suggested by the fact that the corresponding 95% CIs did not include 1.

Table 3 summarizes the effects of TCA on the risk of fracture in subgroup analyses. The risk of fracture was higher among studies focusing on hip/femur fractures (RR, 1.36; 95% CI, 1.26–1.47) when compared to nonvertebral fractures (RR, 1.28; 95% CI, 1.00–1.64). We also found that the risk of fracture was higher when adjusted for BMD when compared to the estimate without a BMD adjustment. Moreover, studies conducted before 2015 (RR, 1.35; 95% CI, 1.19–1.51) and completed outside of the USA (RR, 1.20; 95% CI, 1.00–1.45) had a significant association between TCA use and fracture. The increased risk associated with TCA use was more evident in several subgroups (Table 3), but no significant between-group differences were found (all *p* > 0.15). A moderate to high heterogeneity was observed in most of these subgroup analyses.

## 4. Discussion

The present meta-analysis demonstrated that TCA use is associated with a significantly increased risk of fracture in cohort studies. The association between TCA use and fracture risk was consistent and significant in both frequentist and Bayesian meta-analyses, as well as in all sensitivity analyses and multiple subgroup analyses, which suggested that our findings were robust. Although the fracture risk associated with TCA use found in this meta-analysis was moderate, the estimated absolute risk differences associated with TCA medications could be substantial, given that depression is a prevalent mental illness on a global scale, and that TCAs are still the major prescribed medicines for depressions that have not responded to safer agents or for which safer alternatives are not available [8]. Thus the fracture risk associated with TCA use may have important implications for clinical medicine and public health.

The present study results are consistent with our previous meta-analysis, according to which TCA medication may convey an increased risk of fracture [24]. Our prior meta-analysis included both case-control and cohort studies; however, case-control studies are commonly limited by recall bias and might lead to an inaccurate estimate of the associated effect size. Furthermore, the previous meta-analysis was published eight years ago, and thus it was not able to integrate findings from recently published studies [27,28,29,30,31]. Therefore, our current study focused on the integration of related cohort studies only, and we were able to include four large and well-designed studies [27,28,29,30,31] that have not been included in any prior meta-analyses. We replaced the original study by Lewis et al. [19] in the previous meta-analysis with an updated report by Cauley et al. [30] because both studies used the same data source, and the updated report [23] had a larger sample size and would provide more reliable results.

The findings from the Bayesian meta-analysis were consistent with the results generated from the classical meta-analysis approach. Moreover, the Bayesian meta-analysis provided the probabilities that TCA use increases fracture risk by 10% and 20%. The calculated probabilities at different risk levels offer essential information to physicians in order to help them make clinical intervention decisions regarding the prescription of TCAs for patients. Heretofore, such information was impossible to generate when using a conventional meta-analysis methodology alone.

We observed some variations in the estimated fracture risk associated with TCA use in the subgroup meta-analysis. Fracture risk associated with TCA use in studies that focused on hip/femur fracture outcomes was significantly higher than that in the nonvertebral fracture group. Since hip fracture is more likely to be influenced by falls [24,47], TCAs are highly anticholinergic, often causing sedation and postural instability, both of which appear to increase the likelihood of falls. Although subgroup analyses showed different risk estimates associated with TCA use between the examined subgroups, the estimated risk was not significant in certain groups because of the small number of studies qualified for this meta-analysis. Due to the same reason, the difference between subgroups was not statistically significant in all subgroup analyses. Therefore, additional extensive cohort studies are warranted to investigate whether demographic and clinical variables modify the effect of TCA use on fracture risk. Finally, although our cumulative meta-analysis showed that the pooled estimate fluctuated with time, it showed that since 2011 there has been a consistent and significant association between TCA use and an increased risk of fracture. Additional extensive cohort studies are warranted. The future cumulative meta-analysis should include more eligible cohort studies in order to stabilize the estimate of fracture risk associated with TCA use.

The underlying mechanism of how TCA medication influences fracture risk remains unclear. The increased risk of fracture might be caused by bone loss and a higher propensity to fall. Prior research found a significant association between antidepressant use and an increased risk of falls among the older population [48,49]. TCA medication may cause ataxia, impaired psychomotor function, syncope and additional falls among older adults [50]. In addition, most patients have decreased blood pressure at the initial stage of TCA treatment [51,52], and the risk of falling may increase because of a reduced blood pressure. On the other hand, TCA treatment may also increase patients’ heart rates [53], which leads to an influence on the cardiac output and a decreased blood flow to the central nervous system, then causing confusion and delirium. These adverse effects often lead to an increase in the propensity to fall and thus create an increased fracture risk. 

The adverse effects described above make TCAs an inappropriate psychotropic drug for older patients. The Beers Criteria for Potentially Inappropriate Medication Use in Older Adults and Screening Tool of Older Persons’ potentially inappropriate Prescriptions (STOPP) [50,54] recommended avoiding TCAs in older adults unless safer alternatives were not available. Although TCAs have gradually been replaced by SSRIs and other antidepressants with increased safety, TCA remains a good choice for some patients whose depression has not responded to treatment with less toxic agents [8]. As one of the top choices for pharmacologic interventions in treating clinical depression in the past decades, TCAs have been a more successful treatment for melancholic depression than other antidepressant drug classes [55]. To the best of our knowledge, the present study is the most comprehensive meta-analysis to date to investigate the association between TCA use and fracture risk in cohort studies. Our analysis results from sensitivity analysis, subgroup analysis and cumulative meta-analysis have demonstrated the robustness of the significant association between TCA use and fracture risk. 

There are several limitations associated with our meta-analysis. First, because of the limited number of studies that met the inclusion criteria (*n* = 8), we were not able to perform a multivariate meta-regression analysis to further examine the sources of heterogeneity observed in this meta-analysis. Nonetheless, as described above, we conducted numerous subgroup analyses to determine the variations of the estimated risk by key variables and risk factors. Heterogeneity may be partially caused by subgroup differences in the anatomical site, year of publication, and the study sample and location, as well as by the adjustment for BMD, smoking or osteoporosis. Second, we were not able to assess the risks of falls in this meta-analysis because few eligible original studies accounted for the risk factor of falls. Nonetheless, falls remain an essential potential etiology in the association between TCA use and the risk of fracture. Third, we were not able to assess the effects of different doses of TCA on fracture risk, as the TCA dose was not available in most original studies of this meta-analysis. Fourth, other medications, including glucocorticoids [56,57] and anticonvulsants [58], were also reported to be associated with fracture risk. However, such information was not available in most original studies. Thus we were not able to assess the impact of these confounders on fractures in this meta-analysis. Finally, we approximated HRs as RRs in our current meta-analysis. HR and RR are different, although they are deemed broadly equivalent and have been used interchangeably in much meta-analysis research [59,60,61]. To examine the possible impact on our findings, we conducted a sensitivity analysis that included studies reporting HR only: the corresponding pooled effect size changed slightly and remained significant. Additional research is warranted to examine the role of these factors in the association between TCA treatment and fracture risk.

## 5. Conclusions

Our study suggests that the use of TCAs is associated with an increased risk of fracture. The increased risk associated with TCA use is moderate but may have a substantial clinical impact. Nonetheless, further extensive cohort investigations are warranted, given the variance observed in the subgroup analysis and the unstable estimate found in the cumulative meta-analysis. TCA should be prescribed with caution in the clinic. Fracture risk should be monitored in patients with TCA treatment.

## Figures and Tables

**Figure 1 jcm-09-02584-f001:**
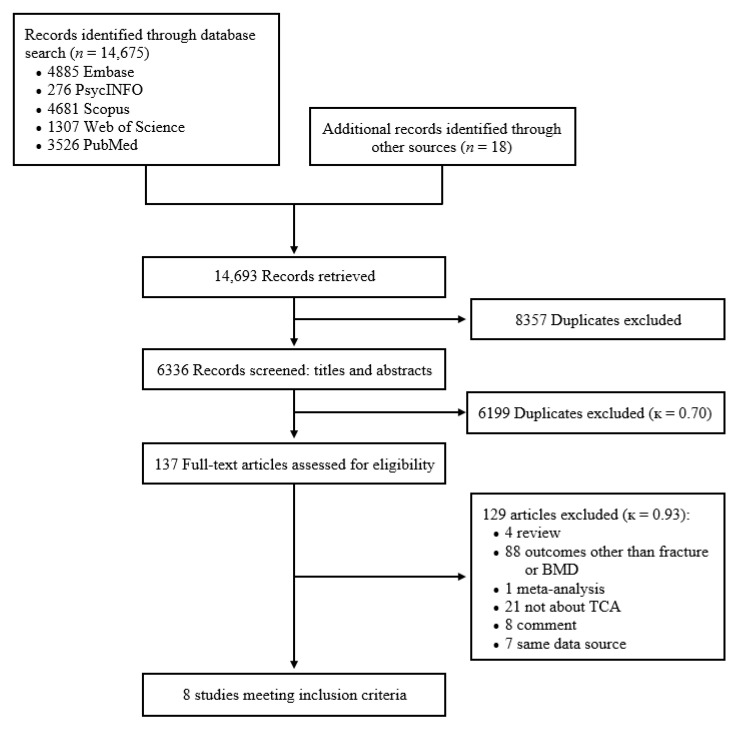
PRISMA flow chart of the study selection (BMD, bone mineral density, TCAs, tricyclic antidepressants).

**Figure 2 jcm-09-02584-f002:**
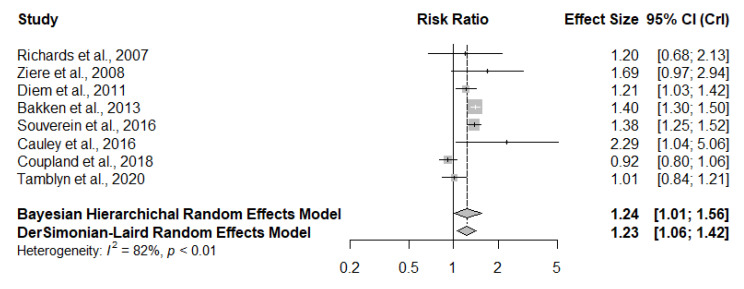
Fracture risk associated with tricyclic antidepressants use for individual studies, and all eligible studies combined by using frequentist and Bayesian approaches. (CI, confidence interval; CrI, Credit Interval).

**Figure 3 jcm-09-02584-f003:**
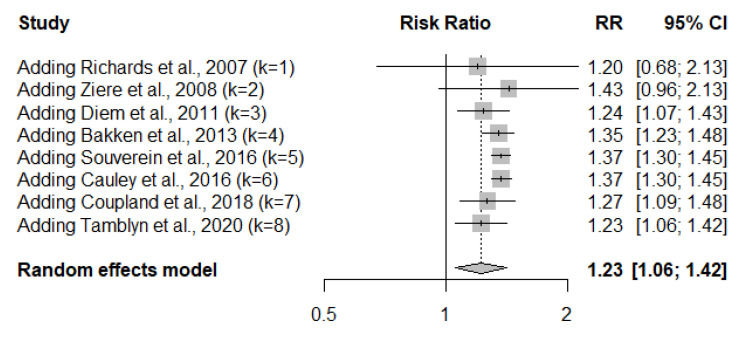
Cumulative random-effects meta-analysis of TCA effects on the risk of fracture. The DerSimonian–Laird method was used for analysis.

**Figure 4 jcm-09-02584-f004:**
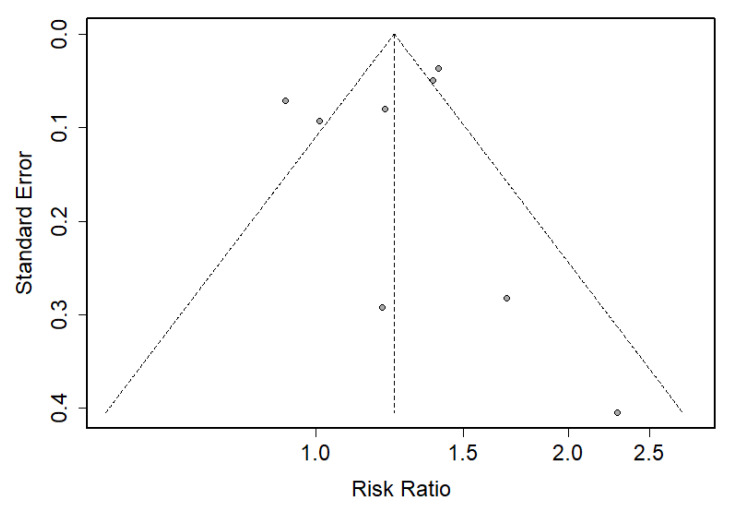
Funnel plot of the risk ratio vs. the standard error. Data were analyzed using the frequentist meta-analysis approach.

**Table 1 jcm-09-02584-t001:** Characteristics of eight cohort studies on the association of tricyclic antidepressants with fracture risk.

Author, Year, Location	Study Population	Exposure Assessment	Outcome Assessment	Outcomes	Mean Follow-Up, Years	Variables Controlled	Funding Source
Richards and colleagues [25], 2007; Canada	The Canadian Multicenter Osteoporosis Study, ≥25 years	Interview	Yearly questionnaire	Fragility fracture	5	Age, total hip BMD, modified Charlson index, prevalent vertebral deformity, prevalent fragility fractures at baseline, cumulative lifetime estrogen use in women.	The Canadian Institutes of Health Research, Merck Frosst Canada Ltd., Eli Lilly Canada Inc, Novartis Pharmaceuticals Inc, The Alliance for Better Bone Health, The Dairy Farmers of Canada, and The Arthritis Society
Ziere and colleagues [26], 2008; Netherlands	Rotterdam Study, ≥55 years	Pharmacy dispensing records	By a medical specialist	Non-vertebral	8.4	Age, sex, depression during follow-up, disability category, and lower-limb disability.	No external funding was obtained for this study
Diem and colleagues [23], 2011; US	Study of osteoporotic fracture, ≥65 years	Interview	Contacted participants about fractures every four months by postcard or telephone, all fractures were confirmed by radiographic reports	Combine fractures of non-spine, first hip, and wrist	10	Age, the status of health, IADL, ability to rise from a chair, m-MMSE, smoking, alcohol drinking, medications including estrogen, bisphosphonate, benzodiazepine, thiazide, PPI and oral steroid medications, body weight, GDS score, walks for exercise, prior fracture history, total-hip BMD, and falls in the previous year.	National Institutes of Health
Bakken and colleagues [29], 2013; Norway	Norwegian Prescription Database (NorPD), no age limitation	The NorPD made assumptions on drug exposure.	Collected national data (i.e., injury, fracture and surgery) on people operated on for hip fracture at all 55 hospitals in Norway performing such surgery	Hip fracture	5.2	Sex, birth year and period (in two-month intervals)	
Souverein and colleagues [31], 2016; Europe	European primary care databases, no age limitation	Prescriptions for antidepressants.	Cases were all patients with a first record/diagnosis of hip/femur fracture identified during follow-up.	Hip/femur fracture	Not specified	Age and sex, previous fracture, systemic glucocorticoid use and rheumatoid arthritis, lifestyle factors (smoking, alcohol use and body mass index; Lifestyle factors were not available for Mondriaan, and information on alcohol use was not available in BIFAP), history of osteoporosis, history of other bone diseases (Paget’s disease and osteogenesis imperfect), previous use of bisphosphonate or any of the other bone protecting drugs.	
Cauley and colleagues [30], 2016; US	MrOS, ≥65 years	Self-report	Hip fractures were verified by a physician using medical records. Pathologic fractures were not included.	Hip	10	Age, race, site and FNBMD, older age (≥75 years), lower FNBMD, currently smoking, fracture after age 50, height and height loss since age 25 years, self-reported history of doctor-diagnosed angina or myocardial infarction, Parkinson’s disease and hyperthyroidism, and poorer executive function.	
Coupland and colleagues [27], 2018; UK	QResearch primary care database, 20–64 years	Prescribed medications	Medical records	Fractures (limb, ribs, skull, vertebrae, pelvis)	5	Age, gender, year of diagnosis of depression, the severity of depression, deprivation, smoking status, alcohol intake, ethnic group, coronary heart disease, hypothyroidism, diabetes, hypertension, cancer, epilepsy/seizures, osteoarthritis, asthma/chronic obstructive airways disease, stroke, osteoporosis, rheumatoid arthritis, renal disease, liver disease, obsessive-compulsive disorder, antihypertensive drugs, anticonvulsants, hypnotics/anxiolytics, statins, NSAIDs, aspirin, hormone replacement therapy, oral contraceptives, antipsychotics, bisphosphonates, anticoagulants, falls.	The National Institute for Health Research (NIHR) School for Primary Care Research
Tamblyn and colleagues [28], 2020; multiple regions	Five jurisdictions in the United States, Canada, The United Kingdom, and Taiwan, ≥65 years	Prescription records	Diagnosis of a fracture from hospitalization databases, medical service procedure for fracture treatment, medical visit for a fracture	Fracture	Not specified	Age, gender, potential indications (depression, anxiety, other mental health issues and pain), conditions that increase fall risk (dementia, Parkinson’s disease, epilepsy, hypertension, peripheral vascular disease, obesity) and fractures risk (cardiac problems, stroke, cancer, renal disease, osteoporosis, fracture history), and concurrent drugs (as time-dependent exposures, benzodiazepines, antipsychotics, and opioids).	The Canadian Institutes of Health Research (CIHR) operating grant MOP-111166

Abbreviations: BMD, bone mineral density; IADL, instrumental activities of daily living; m-MMSE, the Mini-Mental State Examination (modified version); GDS, Geriatric Depression Scale; PPI, proton pump inhibitor; NorPD, Norwegian Prescription Database; BIFAP, Base de datos para la Investigación Farmacoepidemiológica en Atención Primaria; MrOS, Osteoporotic Fractures in Men Study; FNBMD, femur neck bone mineral density; NSAIDs, nonsteroidal anti-inflammatory drugs.

**Table 2 jcm-09-02584-t002:** Risk ratio of fracture associated with treatment of tricyclic antidepressants according to different inclusion criteria.

Studies Included	Number of Studies	Relative Risk (95% CI)	*p*-Value
All studies	8	1.23 (1.06–1.42)	0.0065
All Studies (estimated by Bayesian approach)	8	1.24 (1.01–1.56)	0.02
Studies with TCA use as the primary exposure ^a^	7	1.23 (1.05–1.43)	0.0085
Studies that used HR for risk estimation ^b^	7	1.20 (1.00–1.43)	0.05
Studies that used osteoporotic fractures as outcome ^c^	6	1.37 (1.30–1.45)	0.0001
Studies with participants ≤65 years ^d^	5	1.25 (1.04–1.52)	0.02

The frequentist approach and random-effect model were used for analysis unless noted otherwise. Abbreviations: CI, confidence interval; TCA, tricyclic antidepressant. ^a^ excludes Richards et al. [25] (2007). ^b^ excludes Bakken et al. [29] (2013). ^c^ Excludes Couplant et al. [27] (2018) and Tamblyn et al. [28] (2020). ^d^ excludes Diem et al. [23] (2011), Cauley et al. [30] (2016) and Tamblyn et al. [28] (2020).

**Table 3 jcm-09-02584-t003:** Fracture risk associated with the use of tricyclic antidepressants in the subgroups analysis defined by the following characteristics: anatomical site of fracture, confounder adjustment, years of publication, study location and quality score.

Studies Included	Number of Studies	Risk Ratio (95% CI)	*p*-Value	Between-Group *p*-Value
Anatomical site of a fracture				
Non-spine/Non-vertebral	2	1.28 (1.00–1.64)	0.049	0.59
Hip/Femur	4	1.36 (1.26–1.47)	<0.0001	
Adjusted for BMD				
Yes	3	1.27 (1.01–1.61)	0.04	0.72
No	5	1.2 (1.00–1.45)	0.049	
Adjusted for smoking				
Yes	4	1.22 (0.95–1.55)	0.11	0.84
No	4	1.26 (0.99–1.67)	0.06	
Adjusted for osteoporosis				
Yes	4	1.09 (0.83–1.44)	0.53	0.16
No	4	1.35 (1.22–1.51)	<0.0001	
Year of publication				
<2015	4	1.35 (1.23–1.46)	<0.0001	0.31
≥2015	4	1.16 (0.89–1.53)	0.27	
Location				
USA	2	1.47 (0.83–2.62)	0.19	0.51
International	6	1.20 (1.00–1.45)	0.04	
Quality score				
≤7	5	1.24 (1.05–1.46)	0.009	0.96
>7	3	1.25 (0.86–1.82)	0.23	

No publication bias was observed in this meta-analysis, as indicated by the visual inspection of the funnel plot (Figure 4), and the Begg’s test (*p* = 0.80) indicated that symmetry existed in the funnel plot. Additionally, the use of trim-and-fill correction procedures did not alter the results.

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
