# Peer review of "Tricyclic Antidepressant Use and Risk of Fractures: A Meta-Analysis of Cohort Studies through the Use of Both Frequentist and Bayesian Approaches"

_jcm, 2020, doi:10.3390/jcm9082584_

Round 1
Reviewer 1 Report
The study titled "Tricyclic Antidepressant Use and Risk of Fractures: a Meta-Analysis of Cohort Studies through the use of both Frequentist and Bayesian approaches" is a well conducted and well written Meta-Analysis. The authors are recommended to explain the following points:
- What evidence can the authors cite to support the claim that SSRIs and TCAs are the most commonly prescribed antidepressants? And what kind of geographical restrictions were used to make this claims?
- TCAs are included in the list of potentially inappropriate antidepressant use among older adults based on Beers and STOPP criteria. The authors does not mention about this issue at all. What were the inclusion criteria in terms of age among subjects in the included studies? What are the implications for older adults from this current study (if the authors would like to comment on it)?
- The Introduction section is not very clear in terms of the rationale for conducting this study. The authors mention that the last Meta-Analysis was conducted eight years ago. How has the treatment patterns of TCAs changed during this time period might be more important than the number of studies that were published. Was the intention of this Meta-Analysis simply to update the prior Meta-Analysis findings?
- HR and RR cannot be considered as equivalent. The HR includes a component of time whereas RR does not. So, the basic premise or interpretation of these tow different point estimates are not similar. The authors need to pool the studies reporting HR separately from those reporting RR.
- There are different electronic databases that were used in this study. Was the search strategy same for all the databases or were they tailored (one would expect the latter).
- The authors should consider including the information related to funding sources of the included studies in Table 1.
Author Response
The study titled "Tricyclic Antidepressant Use and Risk of Fractures: a Meta-Analysis of Cohort Studies through the use of both Frequentist and Bayesian approaches" is a well conducted and well written Meta-Analysis.
Authors' response: Thank you for the comments.
What evidence can the authors cite to support the claim that SSRIs and TCAs are the most commonly prescribed antidepressants? And what kind of geographical restrictions were used to make this claims?
Authors' response: Thank you for the comments. In the revision, we revised the contents and acknowledged that the TCAs had been gradually replaced by other safer antidepressant, TCAs are used to treat depression that has not responded to treatment with less toxic agents. We revised the related content in the manuscript. (Please see the 2nd paragraph in the section of Introduction, and the 6th paragraph in the section of Discussion)
TCAs are included in the list of potentially inappropriate antidepressant use among older adults based on Beers and STOPP criteria. The authors does not mention about this issue at all. What were the inclusion criteria in terms of age among subjects in the included studies? What are the implications for older adults from this current study (if the authors would like to comment on it)?
Authors' response: Thank you for the suggestions; we revised the manuscript accordingly. Based on Beers and STOPP criteria, TCA is not an appropriate psychotropic treatment for old people, especially among these who have a history of falls and fractures because TCAs can cause ataxia, impaired psychomotor function, syncope, and additional falls. The adverse effects of TCA partially explain the mechanism of this drug and fracture. (Please see the revision in the 5th and 6th paragraph in the section of Discussion). The age of subjects in the included studies was added in table 1, and briefly described in the section of Results (see 2nd paragraph in the section of Results)
The Introduction section is not very clear in terms of the rationale for conducting this study. The authors mention that the last Meta-Analysis was conducted eight years ago. How has the treatment patterns of TCAs changed during this time period might be more important than the number of studies that were published. Was the intention of this Meta-Analysis simply to update the prior Meta-Analysis findings?
Authors' response: Thank you for the suggestion, and we agree with the reviewer that the change of treatment pattern is an essential and exciting topic. Although TCAs have been gradually replaced by other antidepressants, it is still used to treat depression that has not responded to treatment with less toxic agents. Additionally, with increasing aging populations worldwide, the fracture is a severe clinical and public health concern, which can also lead to significant social and economic burden. The reviewer is correct that the purpose of the current study was to conduct an updated meta-analysis regarding the association between TCA use and fracture risk using both Frequentist and Bayesian approaches. Notably, Bayesian meta-analysis uses the probabilistic method and can help with clinical relevant decision-making when confronted with uncertainty, which has not been used to investigate the association between TCA use and the risk of fracture. We made revision and clarify the aims of this meta-analysis (Please see 1st to 3rd paragraph in the section of Introduction).
HR and RR cannot be considered as equivalent. The HR includes a component of time whereas RR does not. So, the basic premise or interpretation of these tow different point estimates are not similar. The authors need to pool the studies reporting HR separately from those reporting RR.
Authors' response: Thank you for the comments. Based on related literature, HR is broadly equivalent to RR; thus, we approximated HRs as RRs in this study. We revised the contents and added corresponding citations in the revision (see the 2nd paragraph in the subsection of Study Selection, under the 2. Experimental Section). To further address the reviewer's concern, we also conducted an additional analysis to investigate the effect size by HR/RR. Because only one eligible study reported RR, and all other included studies reported HR for risk estimation, we conducted a sensitivity analysis, which focused on studies reporting HR, the corresponding pooled effect size changed slightly. (Please see revision in 4th paragraph and Table 2 in the section of Results)
There are different electronic databases that were used in this study. Was the search strategy same for all the databases or were they tailored (one would expect the latter).
Authors' response: Thank you for the comments. The search strategy was adapted according to the syntax of each specific database. In the revision, we have added the contents and made the clarification. (Please see Search Strategy and Data Sources under 2. Experimental Section).
The authors should consider including the information related to funding sources of the included studies in Table 1.
Authors' response: Thank you for your suggestions. In the revision, we added the information about the funding source of each included study to Table 1. (Please see Table 1)
Reviewer 2 Report
This manuscript takes as its goal to report a metanalysis of all published cohort studies regarding Tricyclic Antidepressant Use and associated risk of fractures. Using Frequentist and Bayesian approaches the apparent underlying objective of the authors appears to help bring to light their opinion that, TCA use is associated indeed with an increased risk of fracture. While the topic is of interest to those treating fractures, I still have some minor revisions that would help improve the quality of the manuscript.
General comment: Authors tend to cite single reference wherever applicable. It would be good if you cite multiple references which can then strengthen your claims.
Specific comments:Line 34: The reference cited here is more than 25year old. please cite recent references.
Line 39-41: It reads, “A widely employed treatment for depression 39 is antidepressant medication, which is one of the most frequently prescribed medicines in Western 40 countries. The most commonly prescribed classes of antidepressants are Tricyclic antidepressants 41 (TCAs) and selective serotonin reuptake inhibitors (SSRIs)”
Please cite the references to supports this statement.
Line 45: In what sense the studies were inconsistent? Please explain
Line 95: Please explain how K-statistic was calculated and cite appropriate references
Line 164: As above
Line 178: It reads, “TCA use increased fracture risk by more than 0%, 10%, and 20% were 98%, 89%, and 63%, respectively.”
This statement is not clear. This statement needs to be formed differently so it is understandable.
Line 220: Please describe the upper limit of RR. At what value of RR you decide that the risk is higher.
Author Response
General comment: Authors tend to cite single reference wherever applicable. It would be good if you cite multiple references which can then strengthen your claims.
Authors' response: Thank you for the suggestion. In the revision, we added additional references when appropriate, especially in the sections of Introduction and Discussion.
Line 34: The reference cited here is more than 25year old. please cite recent references.
Authors' response: Thank you for the suggestion. In the revision, we revised the sentence and replaced the citation with more recent literature. (Please see 1st paragraph in the section of Introduction)
Line 39-41: It reads, "A widely employed treatment for depression 39 is antidepressant medication, which is one of the most frequently prescribed medicines in Western 40 countries. The most commonly prescribed classes of antidepressants are Tricyclic antidepressants 41 (TCAs) and selective serotonin reuptake inhibitors (SSRIs)" Please cite the references to supports this statement.
Authors' response: Thank you for the suggestions. In the revision, we acknowledged that other antidepressants, such as SSRIs, have gradually replaced TCA; however, it is still used to treat depression that has not responded to treatment with less toxic agents. We revised the related contents in the manuscript and added the corresponding references to the contents. (Please see the 2nd paragraph in the section of Introduction)
Line 45: In what sense the studies were inconsistent? Please explain
Authors' response: Thank you for the comments. The "inconsistency" refers to that some studies found a significant association between TCA use and fracture, while others did not. Following the reviewers' recommendations, we revised the contents to clarify in our manuscript. (Please see the 2nd paragraph in the section of Introduction)
Line 95: Please explain how K-statistic was calculated and cite appropriate references
Line 164: As above
Authors' response: Thank you for the recommendation. We added the related content to explain the к statistics and its formula in the revision. (Please see Study Selection under 2.Experimental Section)
Line 178: It reads, "TCA use increased fracture risk by more than 0%, 10%, and 20% were 98%, 89%, and 63%, respectively." This statement is not clear. This statement needs to be formed differently so it is understandable.
Authors' response: Thank you for your comments. We revised this statement in our manuscript. (Please see 3rd paragraph in the section of Results)
Line 220: Please describe the upper limit of RR. At what value of RR you decide that the risk is higher.
Authors' response: Thank you for your comments. We presented both the corresponding effect size (RR), and their upper limits and lower limit of 95% CI of RR in the corresponding lines. In this analysis, we found the effect size of the group focusing on hip/femur fracture was larger than the group focusing on non-vertebral fracture. However, their 95% CIs overlapped, and the p-value for the between-group test was not significant. Thus, the effect size of the group focusing on hip/femur fracture was "higher," however "no significant". These contents have been presented in the 5th paragraph in the section of Results).
Round 2
Reviewer 1 Report
The authors have done a great job addressing my previous comments. However, I am not still convinced about the equivalency of HR and RR (given their basic conceptual differences). I reviewed the Reference #38 cited by the authors but did not really understand how the RR and HR equivalency was established (or if it was really claimed in that way in this study). This is important because, when I look closely to the pooled results from the seven studies reporting HR, the result includes 95%CI of 1-1.43 with a p-value of 0.05. So, the significance is borderline at best. The authors should at least acknowledge that in their limitation sections if not discuss a lit bit on how best to interpret this finding.
Author Response
The authors have done a great job addressing my previous comments. However, I am not still convinced about the equivalency of HR and RR (given their basic conceptual differences). I reviewed the Reference #38 cited by the authors but did not really understand how the RR and HR equivalency was established (or if it was really claimed in that way in this study). This is important because, when I look closely to the pooled results from the seven studies reporting HR, the result includes 95%CI of 1-1.43 with a p-value of 0.05. So, the significance is borderline at best. The authors should at least acknowledge that in their limitation sections if not discuss a lit bit on how best to interpret this finding.
Authors’ response: Thank you for the comments. We agree with the reviewer that there are basic differences between HR and RR. Although hazard ratio (HR) is frequently interpreted as risk ratio (RR), technically, the methods used to calculate HR and RR are different. However, HR is broadly equivalent to RR (https://bestpractice.bmj.com/info/us/toolkit/ebm-tools/a-glossary-of-ebm-terms/). In meta-analysis, HR and RR are often used exchangeable. Many published meta-analysis papers pooled HR and RR together for calculating the pooled effect size, including papers published in the Lancet and JAMA (Ettehad, Dena et al., Blood pressure lowering for prevention of cardiovascular disease and death: a systematic review and meta-analysis, The Lancet, 2015 Dec; 387 (10022), 957-967; Seidelmann SB et al., Dietary carbohydrate intake and mortality: a prospective cohort study and meta-analysis, The Lancet Public Health, 2018 Aug; 3(9):e419-e428; Barone BB et al., Long-term All-Cause Mortality in Cancer Patients With Preexisting Diabetes Mellitus: A Systematic Review and Meta-analysis, JAMA, 2008 Dec; 300(23), 2754-2764). Following the reviewer’s recommendation, in the revision, we acknowledged such limitation in the section of the discussion. (Please see revision in the last paragraph of Discussion)
In conclusion, we believe that these recommendations and the subsequent revisions have strengthened our manuscript substantially. We thank the reviewers again for their helpful suggestions and look forward to the editorial board’s ultimate decision regarding the acceptability of our manuscript.